# Optical Properties of Doxorubicin Hydrochloride Load and Release on Silica Nanoparticle Platform

**DOI:** 10.3390/molecules26133968

**Published:** 2021-06-29

**Authors:** Trong Nghia Nguyen, Thi Thuy Nguyen, Thi Ha Lien Nghiem, Duc Toan Nguyen, Thi Thu Ha Tran, Duong Vu, Thi Bich Ngoc Nguyen, Thi Minh Huyen Nguyen, Van Tien Nguyen, Minh Hue Nguyen

**Affiliations:** 1Institute of Physics, Vietnam Academy of Science and Technology, 10, Dao Tan, Ba Dinh, Hanoi 10000, Vietnam; trongnghia@iop.vast.ac.vn (T.N.N.); thuynt@iop.vast.ac.vn (T.T.N.); ductoan@iop.vast.ac.vn (D.T.N.); thuha@iop.vast.ac.vn (T.T.H.T.); duongvu@iop.vast.ac.vn (D.V.); bichngocphysics53@gmail.com (T.B.N.N.); 2Institute of Bio-Technology, Vietnam Academy of Science and Technology, 18 Hoang Quoc Viet, Cau Giay, Hanoi 10000, Vietnam; nmhuyen09@gmail.com; 3Department of Physics, Graduate University of Science and Technology, Vietnam Academy of Science and Technology, Hanoi 10000, Vietnam; tien85ktqs@gmail.com; 4Department of Physics, Le Quy Don Technical University, Hanoi 10000, Vietnam; huenm293@gmail.com

**Keywords:** silica nanoparticles (SiO_2_ NPs), doxorubicin hydrochloride (DOX·HCl), drug delivery

## Abstract

Silica nanoparticles (SiO_2_ NPs) synthesized by the Stober method were used as drug delivery vehicles. Doxorubicin hydrochloride (DOX·HCl) is a chemo-drug absorbed onto the SiO_2_ NPs surfaces. The DOX·HCl loading onto and release from the SiO_2_ NPs was monitored via UV-VIS and fluorescence spectra. Alternatively, the zeta potential was also used to monitor and evaluate the DOX·HCl loading process. The results showed that nearly 98% of DOX·HCl was effectively loaded onto the SiO_2_ NPs’ surfaces by electrostatic interaction. The pH-dependence of the process wherein DOX·HCl release out of DOX·HCl-SiO_2_ NPs was investigated as well. For comparison, both the free DOX·HCl molecules and DOX·HCl-SiO_2_ NPs were used as the labels for cultured cancer cells. Confocal laser scanning microscopy images showed that the DOX·HCl-SiO_2_ NPs were better delivered to cancer cells which are more acidic than healthy cells. We propose that engineered DOX·HCl-SiO_2_ systems are good candidates for drug delivery and clinical applications.

## 1. Introduction

Doxorubicin (DOX) is an anthracycline antibiotic and is one of the most chemotherapeutic antitumor drugs to treat different solid malignant tumors. DOX consists of a planar anthraquinone nucleus attached to an amino sugar through a glycosidic bond [1]. In order to increase aqueous solubility, the amino group of sugar is protonated by forming DOX hydrochloride (DOX·HCl). The flat anthraquinone part of the molecule is highly lipophilic, while the sugar part is hydrophilic. The DOX·HCl molecule contains various functional groups such as phenolic ring, acid, and base functional groups in the amino sugar. These make the DOX·HCl molecule both amphiphilic and amphoteric [1]. Moreover, DOX·HCl is biocompatible. It interacts with DNA by intercalation to inhibit biomacromolecule synthesis. DOX·HCl is an exciting drug delivery research tool in view of its inherent fluorescence associated with the central anthracycline chromophore group. In recent years, DOX·HCl has been widely used to study cancer therapy employing fluorescence properties to visualize the results. However, free DOX·HCl causes severe side effects such as myelosuppression and cardiotoxicity when treating cancer via chemotherapy [2]. 

Nanotechnology is a promising alternative to overcome these limitations in cancer therapy as it has been shown to reduce the systemic side effects and increase the therapeutic efficacy of drugs [3]. Numerous nanoparticle drug delivery systems, where the chemo-drugs were incorporated inside, have been developed in the last few years. Many types of nanomedicines are currently being investigated in the clinic for cancer treatment, namely proteins drug conjugates, lipid-based, natural or unnatural polymer-based nanocarriers [3,4,5,6,7], viral nanoparticles, and inorganic nanoparticles [8]. SiO_2_ NPs—the most popular NPs—are intensively investigated for biomedical applications. Their potential applications for drug delivery have been investigated and developed at incredible rates [6,8,9,10,11]. SiO_2_ NPs constituents are mainly the siloxane groups that can be broken down via hydrolysis to the orthosilicic acid in the human body after the drug delivery process. These products are biocompatible and are excreted well by the urinary system [11,12].

The main advantages of silica nanostructures in biological applications are that they can serve as protection layers or drug delivery vehicles, given that silica in biological media is highly stable. The silica shells possessing no cytotoxicity and outstanding biocompatibility improve the stability of the protected substances in the cores significantly.

Due to the silanols groups (Si-OH) on the pores and surfaces of SiO_2_ NPs, which confer a very negative zeta potential, the positively charged drugs can be absorbed onto SiO_2_ NPs pores and surfaces easily by the strong electrostatic interaction [13]. More importantly, knowing this mechanism, one can control the drug-releasing process at the targets [14]. Finally, it is worth mentioning that silica shells can alter the optical properties of the contained dye molecules—in our case, is DOX·HCl. 

There are two main types of SiO_2_ NPs: nonporous and mesoporous silica, which can be obtained from the hydrolysis and condensation of silane alkoxide precursors. The large pore of mesoporous silica nanoparticles can shelter large biomolecules such as proteins and DNA, protecting them from rapid biodegradation. Therefore, many works have focused on drug load onto the mesoporous silica nanoparticles [6,7,9,10,15], which were traditionally created by micelle methods using cationic or anionic template agents. However, these synthesis methods require the use of either surfactant agents or/and toxic solvents, which require great care before applying. Moreover, the nonporous SiO_2_ NPs can be easily synthesized in non-toxic media, nonsurfactant agents with a precise size-control in the range of 10 to 2000 nm from TEOS by the Stober method [16,17,18]. Furthermore, the encapsulation and conjugation of drugs inside these particles have been reported in [19], where authors realized covalent binding between the drugs and the organic functional silane alkoxide precursors during the SiO_2_ NPs formation, imposing the drug inside the silica matrix via the formation of bonds between the drug and silane coupling reagent. However, to the best of our knowledge, the adsorption of the drugs onto SiO_2_ NPs’ surfaces via non-covalent binding after SiO_2_ NPs have been formed, and its uses for drug delivery vehicles as a controllable drug adsorption/release mechanism has not been reported. 

In this work, the optical properties of the free DOX·HCl molecule and of the DOX·HCl adsorbed SiO_2_ NPs after NPs formation were investigated. The drug loading efficiency was monitored using absorption spectra and zeta potentials. The results showed that nearly 98% of DOX was effectively loaded onto the SiO_2_ NPs surface by electrostatic interaction. The DOX release from these NPs, depending on the pH of the solution, was simultaneously measured by fluorescence spectra. The free DOX·HCl molecule and DOX·HCl-SiO_2_ NPs were used to label cultured cancer cells. Confocal laser scanning microscopy (LCSM) images showed that the DOX·HCl-SiO_2_ NPs were efficiently delivered to cancer cells. These results show that the DOX·HCl-SiO_2_ NPs system is an effective method of drug delivery and has great potential for clinical applications.

## 2. Materials and Methods

### 2.1. Materials

Tetraethylorthosilicate (TEOS, 98%) and ammonium hydroxide (NH_4_OH, 28–30% in water), penicillin/streptomycin (P/S), and FBS were purchased from Sigma-Aldrich (St. Louis, MO, USA). Ethanol, hydrochloric acid (HCl, 36.5%), trisodium citrate dehydrate (Na_3_C_6_H_5_O_7_.2H_2_O, 99%), potassium carbonate (K_2_CO_3_, 99%), and tris(hydroxymethyl)aminomethane (C_4_H_11_NO_3_ 99%) were purchased from Merck (Kenilworth, NJ, USA). Doxorubicin hydrochloride solution (2 mg/mL) was obtained from Ebewe Pharma (Unterach am Attersee, Austria). The DMEM medium was purchased from HIMEDIA (Mumbai, India). Deionized water was used in all experiments. 

### 2.2. Cell Culture and Cellular Uptake Assay

The MCF7 breast cancer cell line was maintained in DMEM medium supplemented with 1% penicillin/streptomycin (P/S) and 10% FBS. The cells were incubated at 37 °C under 5% CO_2_ atmosphere. After thawing, the cells were cultured and passaged 2 or 3 times before performed the cellular uptake assay. For treatment with NPs, cells were counted and aliquoted at a concentration of 5000 cells/mL and 1 mL/well in a 12-well-plate. The plate was incubated overnight for cell attachment. The nucleus of MCF7 cells was stained with the DAPI molecules. Then the cells were treated with 4 µM of DOX·HCl, DOX·HCl-SiO_2_ NPs, and the samples were observed after staining 2 and 6 h using confocal laser scanning microscopy. 

### 2.3. Silica Nanoparticles Preparation

#### 2.3.1. Synthesis of SiO_2_ NPs 

SiO_2_ NPs with the sizes of 90–100 nm were prepared by an effortless method. Briefly, 2.0 mL of NH_4_OH were added into 30 mL of ethanol and thoroughly mixed by magnetic stirrer at room temperature for 30 min. After that, 300 µL of TEOS was dropped into the reaction mixture and kept for one day under vigorous stirring. The obtained particles were then washed three times via centrifugation with DI water and redispersed in DI water before DOX·HCl loading.

#### 2.3.2. Doxorubicin Loading onto the SiO_2_ NPs Surface 

In order to determine the maximum loading capacity of DOX·HCl onto SiO_2_ NPs’ surfaces, different volumes of DOX·HCl solution (2 mg/mL) varying from 0 to 60 µL was added into 1 mL of SiO_2_ NPs suspension. These mixtures were incubated and shaken gently for 3 h to allow DOX·HCl to get absorbed onto the SiO_2_ NPs surfaces. The free DOX·HCl molecules and DOX·HCl absorbed on the SiO_2_ NPs (DOX·HCl-SiO_2_) were separated using centrifugation. The supernatant was removed from the pellet with a pipette. After that, the pellet was washed twice again and redispersed in DI water. The stock solution of DOX·HCl-SiO_2_ NPs in aqueous was kept in the dark and at low temperature (about 4–8 °C) for future use. The concentration of the residual DOX·HCl in the supernatant was determined by measuring the absorbance. The following equation is used to calculate the DOX·HCl absorption efficiency (*EF*).
(1)$$EF\left(\%\right)=\left(\frac{L_{i}-L_{f}}{L_{i}}\right)\times 100$$
where $L_{i}$ is the initial DOX·HCl amount used to load on the SiO_2_ NPs and $L_{f}$ is the residual DOX·HCl amount in the supernatant.

#### 2.3.3. pH-Dependence of Doxorubicin Release from SiO_2_ NPs 

The following experiment was conducted to determine the rate at which DOX·HCl molecules go off the SiO_2_ NPs’ surface and how the pH affects this rate. The equal amounts of stored DOX·HCl-SiO_2_ NPs solution were diluted 20 folds in the different pH buffers from 3.0 to 8.0 at an optical density below 0.1. The DOX·HCl content loaded on SiO_2_ NPs is about 1.5 µg·mL^−1^. The released DOX·HCl concentration in the supernatant solution was determined by fluorescence spectra. The concentration of the released DOX·HCl was estimated when compare the measured fluorescent spectra with that of the known concentration of free DOX·HCl in an aqueous solution.

### 2.4. Characterization

The absorption spectra were recorded on a Shimadzu UV-2600 spectrophotometer. The fluorescent spectra were measured on Cary Eclipse fluorescence spectrophotometer. The size, shape, and morphology of SiO_2_ NPs and DOX·HCl-SiO_2_ NPs were observed through TEM Model Jem JEOL 1010 microscope. The zeta potential values were determined by the dynamic light scattering (DLS) technique (Nano ZS. Malvern). Nikon Laser Scan Microscope TE was used to acquire confocal photoluminescence images. The 488 nm, 2.5 mW laser beam was used to excite the fluorescence of the DOX·HCl. The wavelength of the laser is close to the maximum value of the excitation spectrum of DOX·HCl. 

## 3. Results and Discussion

### 3.1. Optical Characterizations of DOX·HCl Molecule

#### 3.1.1. Optical Properties of Free DOX·HCl in Aqueous Solution

The maximum absorption and fluorescence spectra of free DOX·HCl in an aqueous solution as a function of concentration in the range of 1.72 × 10^−6^ M to 1.72 × 10^−5^ M or 1 µg/mL to 10 µg/mL were presented in Figure 1a,b, respectively. This result showed that in this concentration range, the absorbance increases linearly with DOX·HCl’s concentrations. The molar absorption coefficient of DOX·HCl in water at 480 nm is estimated to be 13,500 ± 100 L mol^−1^ cm^−1^. Figure 1b shows that the fluorescence intensity at 595 nm is nearly linear when the DOX·HCl ‘s concentrations are lower than 1 × 10^−5^ M or equivalent to 5 µg/mL.

Quantum yield (*Q.Y.*) of free DOX·HCl in aqueous solutions was determined from the data presented in Figure 2 for both absorption and fluorescence measurements. At this concentration range, the OD varies from 0.01 to 0.15. Solution of R6G dye in ethanol was used as the reference sample. From the known value *Q.Y.* of 94% of the R6G, *Q.Y.* of free DOX was calculated using the following equation.
(2)$$Q.Y.=Q_{F}=Q_{RF}\frac{n^{2}}{n_{R}^{2}}\frac{\int_{0}^{\infty }I_{F}(\lambda_{F})d\lambda_{F}}{\int_{0}^{\infty }I_{RF}(\lambda_{RF})d\lambda_{RF}}\frac{1-10_{RF}^{-O.D.}}{1-10_{F}^{-O.D.}}$$
where $Q_{RF}$ is the *Q.Y.* of R6G molecule in ethanol, $n_{R},\;n$ are the refractive indexes of the ethanol and water corresponding. The $1-10_{F}^{-O.D.}\;$ and $1-10_{F}^{-O.D.}$ terms are the absorption of the sample and reference at the excitation wavelength (488 nm). $I_{F}$ and $I_{RF}$ are the corrected fluorescence spectra of the sample and reference, respectively. The integrals represent the area under the fluorescence spectra. The determined *Q.Y.* of free DOX·HCl in aqueous solution is 4.5% which agrees well with Shah’s report [20]. 

#### 3.1.2. Characterization of DOX·HCl in Different pH Media

The absorption and fluorescence spectra of the DOX·HCl in different buffer solutions with pH varying from 3.0 to 8.0, but the concentration of DOX·HCl was kept constant at 6.8 µM, showed that their intensity and spectral shape have no significant changes when the pH value of buffer solution below 7.2. However, at pH is 8.0 or higher, their absorbances strongly decreased, and the absorption band broadens (Figure 2a). At the same time, Figure 2b shows that their fluorescence intensities are strongly quenched when pH is higher than 8.0. DOX·HCl was found extremely unstable to alkaline hydrolysis, even at room [21]. This explains the substantial decrease of absorbance and intensities of fluorescence spectra of the DOX·HCl when pH is higher than 8.0.

### 3.2. DOX·HCl-SiO_2_ NPs Characterizations 

#### 3.2.1. Zeta Potential of DOX·HCl Loading on the SiO_2_ NPs

Table 1 shows zeta potential values of the SiO_2_ NPs loaded with different amounts of DOX·HCl. The zeta potential value of SiO_2_ NPs in an aqueous solution is highly negatively charged −50 mV while zeta potential of free DOX·HCl is only −27.4 mV. The zeta potential of non loaded SiO_2_·NPs was −50 mV, whereas the zeta potential of SiO_2_ NPs increases with the amount of the DOX·HCl loaded on their surfaces (shown in Table 1). This potential reaches the value of the free DOX·HCL when DOX·HCl molecules fully cover the SiO_2_ NPs surfaces with 120 µg/mL. Based on the measured zeta potential results, we can determine the DOX·HCl loading capacity of the SiO_2_ NPs in an aqueous solution. 

Figure 3A is the bottle containing SiO_2_ NPs suspension before DOX·HCl loading; the SiO_2_ NPs are well dispersed in water. In comparison, Figure 3B,C show the freshly prepared and stay-for-some-time DOX·HCl-SiO_2_ NPs suspensions, respectively. All the DOX·HCl-SiO_2_ NPs subside and lay at the bottom of the bottle, making the top solution nearly transparent. These images obviously indicate that DOX·HCl molecules are totally loaded onto the SiO_2_ NPs’surfaces, and no free DOX·HCl molecules are left in the solution. The results presented in Table 1 indicate that the zeta potentials of the DOX·HCl-SiO_2_ NPs are below −30 mV. Usually, DOX·HCl-SiO_2_ NPs whose zeta potential is less than −30 mV are considered stable. However, The DOX·HCl-SiO_2_-120 NPs are not stable, even though they are dispersed in the medium. The instability of the DOX·HCl-SiO_2_-NPs-120 system in DI water can be treated with bovine serum albumin protein coating the DOX·HCl-SiO_2_-NPs to form the fort steric stabilization [22,23,24].

#### 3.2.2. Morphology of SiO_2_ NPs and DOX·HCl-SiO_2_ NPs

Figure 4A,B show the TEM images of the bare SiO_2_ NPs and DOX·HCl-SiO_2_ NPs. The bare SiO_2_ NPs have a spherical shape with an average diameter 90 ± 5 nm. The diameters of the SiO_2_ NPs stay nearly the same after DOX·HCl loading. The TEM images of the DOX·HCl-SiO_2_ NPs are darker than the ones of the bare SiO_2_ NPs. This higher contrast is due to the absorbed DOX·HCl molecules onto the surfaces of the SiO_2_ NPs. This result is in good agreement with Min Zhang, and Li Jiang’s work reported in 2016 when they load DOX·HCl onto the mesoporous SiO_2_ NPs [4].

#### 3.2.3. Optical Properties of DOX·HCl-SiO_2_ NPs 

The loading capacity of SiO_2_ NPs is evaluated by measuring DOX·HCl concentration in solution before and after loading into SiO_2_ NPs using absorbance measured at 480 nm. Figure 5a,b shows the absorption spectra of DOX·HCl bond on the SiO_2_ NPs and excessive DOX·HCl in the supernatant compared to those of free DOX·HCl in an aqueous solution. The absorption band of the DOX·HCl bond on SiO_2_ NPs is broader than those of free DOX·HCl in an aqueous solution (Figure 5). Moreover, the molar absorption coefficient at 480 nm of DOX·HCl-SiO_2_ NPs decrease three folds. The silanol group on the SiO_2_ NPs attracts the daunosamine head of the DOX·HCl molecule via electrostatic force. Therefore, DOX·HCl molecules can anchor onto the surface of the SiO_2_ NPs, making the flat anthraquinone head sticks outwards from the NPs’ surface. This alignment of the DOX·HCl molecules make the absorption cross-section of the DOX·HCl molecules changed.

Figure 5b compares the spectra of DOX·HCl in the supernatant and free DOX·HCl at a 2 µg/mL concentration. Their absorbances at 488 nm are roughly the same. Based on this, we can estimate that the DOX·HCl concentration in the supernatant is close to 2 µg/mL and hence quantify the amount of DOX·HCl absorbed on the SiO_2_ NPs. The loading efficiency of DOX·HCl molecules into SiO_2_·NPs can reach up to 98%. For the 90 ± 5 nm NPs, the estimated loading content is 30 µg of DOX·HCl per mg of SiO_2_ NPs. Reducing the sizes of the NPs, makes the surface area/volume increase. As a consequence, the DOX·HCl storage of SiO_2_ NPs becomes bigger.

Figure 5c presents the fluorescence spectra of 120 µg DOX·HCl binding with 1 mL SiO_2_ NPs suspension, excess DOX·HCl content in the supernatant and DOX·HCl in water at 2 µg/mL. It is clear that the fluorescence of DOX·HCl-SiO_2_ NPs is weak, while those of the supernatant and 2 µg/mL DOX·HCl solution in water are comparable. Especially, this result demonstrates that free DOX·HCl molecules fluoresce much better than DOX·HCl bound to SiO_2_ NPs. 

#### 3.2.4. pH-Dependence of DOX·HCl Release from DOX·HCl-SiO_2_ NPs 

As mentioned above, DOX·HCl-SiO_2_ NPs do not fluoresce well while free DOX·HCl molecules in an aqueous solution fluoresce with a *Q.Y*. of 4.5%. Therefore, we can observe the DOX·HCl release via fluorescent measurement. Figure 6 shows the fluorescence intensities of DOX·HCl-SiO_2_ NPs in different buffer solutions as a function of time. These curves show the efficiencies of DOX·HCl release from DOX·HCl-SiO_2_ NPs in the different buffers having pH range from 3.0 to 8.0. This result reveals that in the water or the buffer with pH higher than 7.25 the DOX·HCl gets off DOX·HCl-SiO_2_ NPs very slowly. We recall that the pH values of the blood and physiologic media are around 7.25–7.50. In the first 5 h, the release rate increase with decreasing pH. The curves go flat after 24 h, hinting that there is almost no release afterward (shown in Figure 6). In comparison, the accumulative content of DOX·HCl release attains 90% at pH = 3.0 and 50% at pH = 5. These pH values are similar to the extracellular pH of tumors and in the liposome of cellular [5]. This pH dependence is very interesting. It allows us to realize the drug delivery schema so that DOX·HCl-SiO_2_ NPs are stable in the blood and follow the circulation system to reach the targeted tumors. At the tumor tissues, DOX·HCl molecules release from the DOX·HCl-SiO_2_ NPs and attack them.

#### 3.2.5. Intracellular Uptake of DOX·HCl-SiO_2_ NPs

The cellular uptake of DOX·HCl and DOX·HCl-SiO_2_ NPs was further studied with MCF7 cells. DAPI was used for nuclear staining and the DOX·HCl fluorescence was monitored in cells by a laser scanning confocal microscope. Figure 7 shows the fluorescent images of DOX·HCl and DOX·HCl-SiO_2_ NPs treated cells. After treatment with DOX·HCl-SiO_2_ NPs, the intracellular delivery was significantly enhanced compared with cells treated with DOX·HCl molecules. An apparent increase in the amount of drug passed through the cell membrane was obviously observed after two hours of incubation with DOX·HCl. Cellular uptake and the LCSM analyses showed that the free DOX·HCl started to enter the nuclei of MCF7 cells after two hours and totally concentrated in the nuclei after six hours. In contrast, after two hours of incubation with DOX·HCl-SiO_2_ NPs system, the nuclear almost have the DAPI‘s color. However, after six hours of incubation with DOX·HCl-SiO_2_ NPs, the similar results with the free DOX·HCl, the superposing of the red color with the blue color in the nuclear of MCF7 cells, was observed. This phenomenon can be explained that the pH media around cells is close to pH 7.4 of DMEM medium in the early time. Over time, the pH of media becomes more acidic [25,26], increasing the release of DOX·HCl from the DOX·HCl-SiO_2_·NPs. This result agrees with Guan‘s report where authors observed that after two hours of incubation, cells with DOX·HCl or their DOX·HCl nanocarrier system, DOX·HCl was found in the nuclear [5], especially in acidic media at pH of 6.8. These results demonstrated that the acid-sensitivity property of the DOX·HCl-SiO_2_ NPs boosted the DOX·HCl release in the tumors, which led to a significant killing effect on the cancer cells. More importantly, the system showed remarkable cytotoxicity against MCF7 cancer cells. In our experiment, Figure 7 showed the positions where DOX·HCl accumulates and the positions of DAPI in the nucleus of the MCF7 cancer cells. These results confirmed that DOX·HCl molecules release from DOX·HCl-SiO_2_ NPs inside cells and the DOX·HCl molecules bind to DNA which is the main cause of cells’ death [2,27,28].

## 4. Conclusions

DOX·HCl-SiO_2_ NPs is a drug delivery system that exhibited dual functions. The zeta potential analysis confirms the charge change of DOX·HCl-SiO_2_ NPs complexes to determine the maximum DOX·HCl loading. The optical characterization was used to monitor the loading and release of DOX·HCl on the DOX·HCl-SiO_2_ NPs. CLSM results showed that the DOX·HCl released from DOX·HCl-SiO_2_ NPs was delivered efficiently to the nucleus of cancer cells. These results show that the DOX·HCl-SiO_2_ NPs system is a promising candidate for drug delivery and has potential in clinical applications.

## Figures and Tables

**Figure 1 molecules-26-03968-f001:**
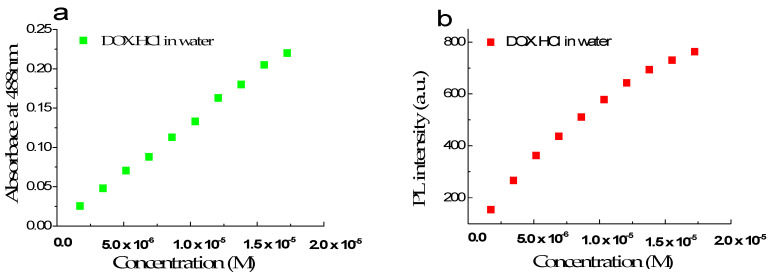
Absorption and fluorescence intensity spectra of free DOX in aqueous solution at 480 nm (**a**) and 595 nm (**b**) with the concentration in the range of 1.72 × 10^−6^ M to 1.72 × 10^−5^ M.

**Figure 2 molecules-26-03968-f002:**
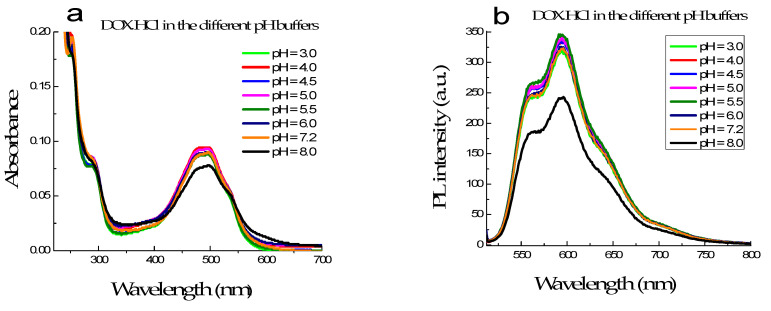
The absorption (**a**) and fluorescence (**b**) spectra of DOX·HCl in varying pH buffer solution from 3.0 to 8.0.

**Figure 3 molecules-26-03968-f003:**
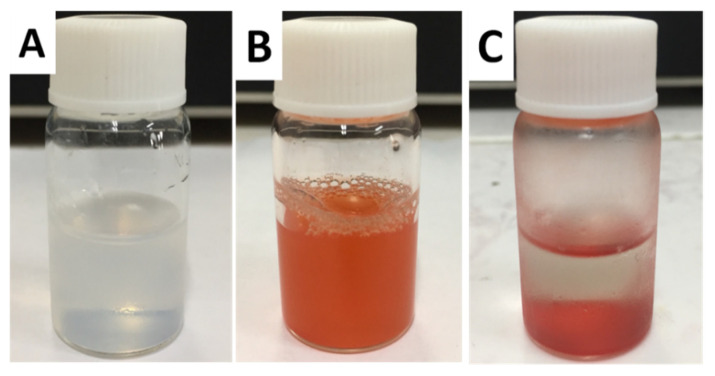
The SiO_2_ NPs solutions before (**A**); freshly prepared and stay-for-some-time DOX·HCl-SiO_2_ NPs (**B**,**C**).

**Figure 4 molecules-26-03968-f004:**
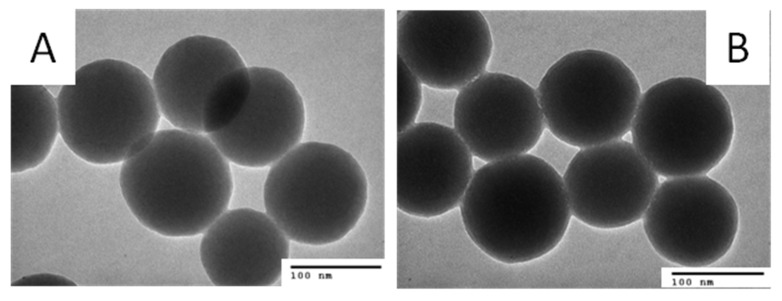
TEM images of SiO_2_·NPs at scale bare of 100 nm before (**A**) and after (**B**) absorbed DOX·HCl molecules.

**Figure 5 molecules-26-03968-f005:**
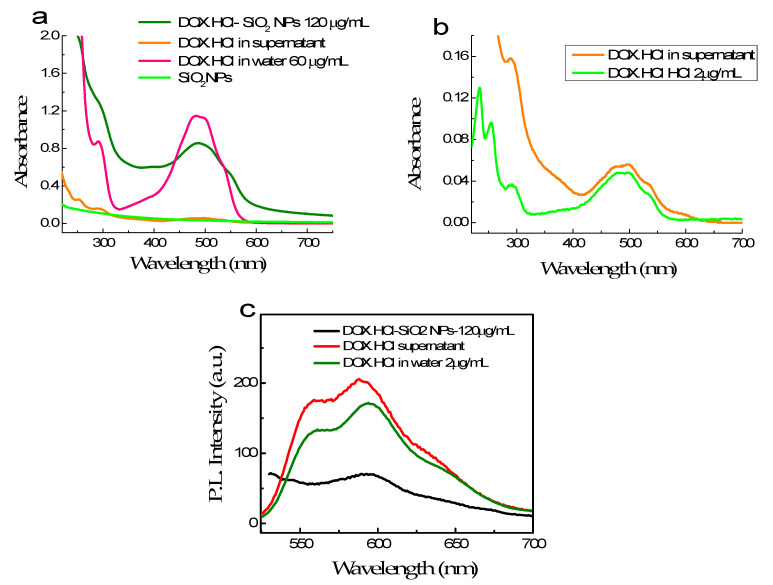
The absorption spectra of SiO_2_ NPs, 120 µg/mL DOX·HCl absorbing on SiO_2_ NPs and residual DOX·HCl content in supernatant compared with those of free DOX·HCl at 60 µg/mL, and the residual DOX·HCl in the supernatant (**a**); the absorption of DOX·HCl in supernatant compared with those of free DOX·HCl at 2 µg/mL in aqueous solution (**b**); and the fluorescence spectra of three above solutions (**c**).

**Figure 6 molecules-26-03968-f006:**
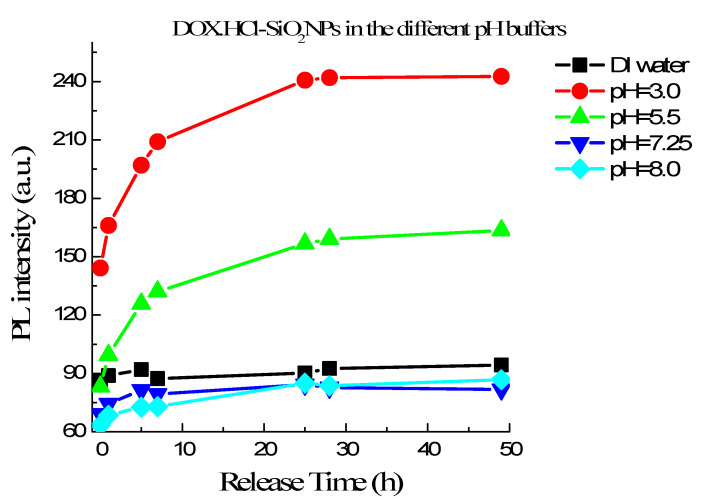
DOX·HCl release profiles from DOX·HCl-SiO_2_ NPs in the different pH buffers varying from 3.0 to 8.0 in time.

**Figure 7 molecules-26-03968-f007:**
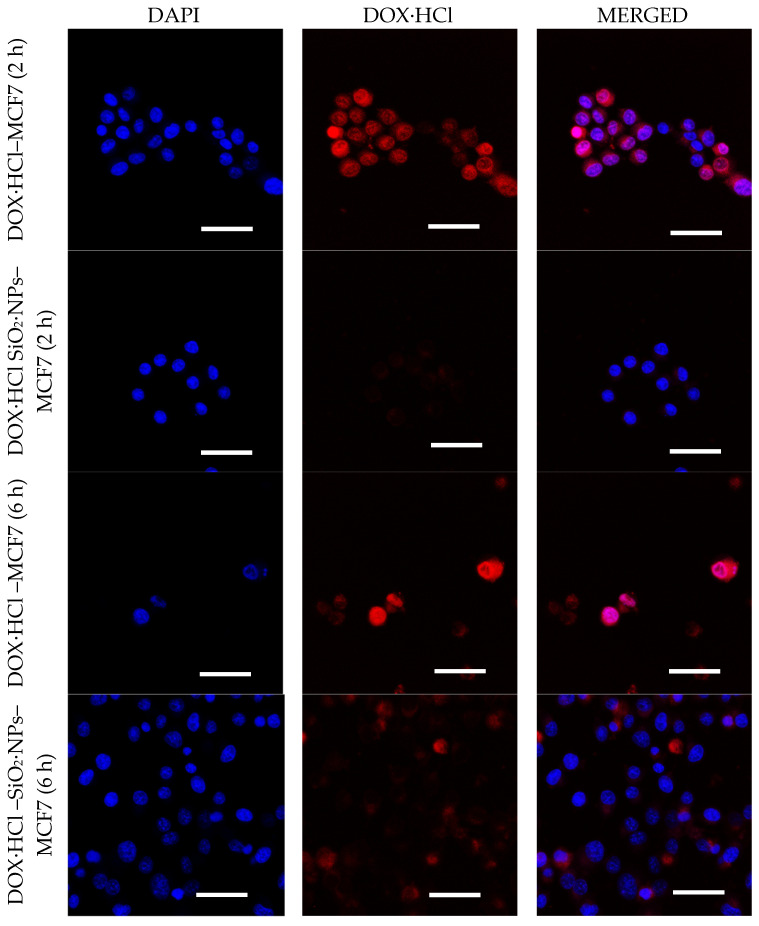
CLSM images of MCF7 cells incubated with DOX·HCL molecule (on top), DOX·HCl–SiO_2_ NPs system (bottom) at DOX·HCl concentration of 4 µM. Images were acquired after 2 and 6 h. (Nuclei of these cells incubated with DAPI molecules). Scale bar 50 μm.

**Table 1 molecules-26-03968-t001:** Zeta potential values of bare SiO_2_ NPs and different quantity DOX·HCl absorbed on SiO_2_ NPs in an aqueous solution.

Sample	DOX·HCl Loading (µg/mL)	Zeta Potential (mV)
SiO_2_ NPs	0	−50.0
SiO_2_–DOX·HCl-20	20	−50.3
SiO_2_–DOX·HCl-60	60	−42.2
SiO_2_–DOX·HCl-120	120	−35.4
DOX. HCl	20	−27.4

## Data Availability

The data presented in this article are available on request from the corresponding authors.

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
