# Peer review of "Optical Properties of Doxorubicin Hydrochloride Load and Release on Silica Nanoparticle Platform"

_molecules, 2021, doi:10.3390/molecules26133968_

Round 1
Reviewer 1 Report
The manuscript of Nguyen et al report the loading, release and properties of doxorubicin in silica nanoparticles. This has been profusely reported in the literature and I can not see which is the novelty or significance of these results. Please, consider to improve the introduction and highlight which is the novelty of this work in relation to all the previous literature about dox-loaded SiO2NPs for drug delivery and as antitumoral agents
Author Response
Dear Mr./ Ms. Reviewer,
We want to thank you for reviewing our manuscript. We appreciate your constructive comments. We have improved the introduction and highlight which is the novelty of the work in relation to all the previous literature about DOX loaded SiO2 NPs for drug delivery and as antitumor agents.
Sincerely,
Reviewer 2 Report
The article "Optical Properties of Doxorubicin Hydrochloride Load and Release on Silica Nanoparticle Platform" shows the obtaining and characterisation of Doxorubicin-loaded SiO2 nanoparticles for drug delivery applications.
Although the manuscript is well written, major improvements should be done before being considered for publishing. Please, find my suggestions below:
Abstract: Lines 22-23- the statement is not supported by the results. Although the characterisation is promising, it is not sufficient to draw such conclusion. This should be revised.
Introduction: There are a lot of affirmations lacking literature references, for example lines 53-56 or 56-57. This should be sustained by references.
lines 64-66- not relevant for the manuscript and the context. Other more relevant example should be considered (one involving SiO2- based platforms).
The subject presented is not new. Thus, the introduction should clearly outline the purpose of the study and also the novelty which is brought by the manuscript.
Materials and methods: The synthesis and drug loading should be describes in more detail and any novelty in the obtaining of the material should be stated.
The DOX releasing procedure should be describe more concisely.
Results and discussion: The Zeta Potential measurements after DOX loading show a high stability, respectively a good stability in water. However, the pictures in figure 3 reveal that the loaded SiO2 have lost their stability over time. How can you explain this? Moreover, the sample meaning in table 1 is not given. This should be detailed either in the table description, either in Materials and Methods section.
Line 196 lack of references.
Overall, I suggest that the discussion part of the manuscript should be sustained more with data from the literature and the novelty of the manuscript should be emphasised.
Cell death over time should be measured using at least one appropriate assay, such as the MTT assay.
Figure 7- the images are very low quality. Clearer images should be provided. Also, images at higher magnification could be helpful.
Conclusions: Again, the statement regarding the possible clinical applications of the constructs is not sustained by the data presented in the manuscript.
References: Very few and outdated references used. Newer data should be referred in the state of the art of the manuscript (articles from 2017-2016 the earliest).
Author Response
Dear Mr./Ms. Reviewer,
Dear Mr./Ms. Reviewer 2.
We want to thank you for reviewing our manuscript. We appreciate your constructive comments. We have addressed the feedback and improved our manuscript. Details of our response as following:
The article "Optical Properties of Doxorubicin Hydrochloride Load and Release on Silica Nanoparticle Platform" shows the obtaining and characterization of Doxorubicin-loaded SiO2 nanoparticles for drug delivery applications.
Although the manuscript is well written, major improvements should be done before being considered for publication. Please, find my suggestions below:
Point 1: Abstract: Lines 22-23- the statement is not supported by the results. Although the characterization is promising, it is not sufficient to draw such conclusion. This should be revised.
Response 1: According to the obtained results on the release of DOX.HCl-SiO2 NPs depend on the pH and based on the acidic extracellular microenvironment of tumor or cancer cells was formed overtime, we have revised this point “DOX.HCl-SiO2 NPs were better delivered to cancer cells which are more acidic than healthy cells”
Point 2: Introduction: There are a lot of affirmations lacking literature references, for example lines 53-56 or 56-57. This should be sustained by references.
Response 2: We appreciate your constructive comments for the introduction. We have added the lacking literature references for examples lines 53-56. The references for example on lines 56-57, we have cited works done by Croissant group in 2017, they reported on the degradability and clearance the nanoparticles were formed from silicon and silicon dioxide materials (reference number [11])
Point 3: lines 64-66- not relevant for the manuscript and the context. Other more relevant example should be considered (one involving SiO2- based platforms).
Response 3: We want to thank to reviewer for this remark, we removed these lines from the manuscript, and more relevant example of SiO2 NPs based platform has been added in the introduction.
Point 4: The subject presented is not new. Thus, the introduction should clearly outline the purpose of the study and also the novelty which is brought by the manuscript.
Response 4: We want to thank you for reviewing our manuscript. We appreciate your constructive comments. We have improved the introduction and highlight which is the novelty of the work in relation to all the previous literature about DOX loaded SiO2 NPs for drug delivery and clearly outline the purpose of study in our manuscript.
Point 5: Materials and methods: The synthesis and drug loading should be describes in more detail and any novelty in the obtaining of the material should be stated.
The DOX releasing procedure should be describe more concisely.
Response 5: The novelty in the obtaining of the material has been described in the introduction, and the DOX.HCl-SiO2 NPs releasing procedure has been described more concisely.
Point 6: Results and discussion: The Zeta Potential measurements after DOX loading show a high stability, respectively a good stability in water. However, the pictures in figure 3 reveal that the loaded SiO2 have lost their stability over time. How can you explain this? Moreover, the sample meaning in table 1 is not given. This should be detailed either in the table description, either in Materials and Methods section.
Response 6: We want to thank reviewer. We appreciate your constructive comments.
The stability of a colloidal system is determined by the sum of the van der Waals attractive, electrical double layer repulsive forces. The balance between the attractive and repulsive force is important for the stability of the colloidal system in the suspensions. In the zeta potential measurement after DOX.HCl loading, the obtained results show the zeta potentials of the colloidal systems are below -30 mV. Usually, DOX.HCl-SiO2 NPs whose zeta potential is less than -30 mV are considered stable. The DOX.HCl-SiO2-120 NPs are not stable, even though they are dispersed in the medium. Howerer, the instability of the DOX.HCl-SiO2-NPs-120 system in DI water can be treated with bovine serum albumin protein coating the DOX.HCl-SiO2-NPs to form the fort steric stabilization.
Point 7: Line 196 lack of references
Response: The lack of references in line 196 has been cited by work of Min Zhang and Li Jiang’s group done in 2016 with reference number [4].
Point 8: Overall, I suggest that the discussion part of the manuscript should be sustained more with data from the literature and the novelty of the manuscript should be emphasised.
Response 8: We want to thank reviewer for your constructive suggestions. In the discussion part of the manuscript have been sustained with data from literature. And the novelty of the manuscript have been emphasised.
Point 9: Cell death overtime should be measured using at least one appropriate assay, such as the MTT assay
Response 9: DOX.HCl is biocompatible, it interacts with DNA by intercalation to inhibit biomacromolecule synthesis, cause cells’ death over time. Thus, the present of the DOX.HCl molecules inside the cell binding to DNA in nucleus when using the DOX.HCl-SiO2 NPs obviously observed via fluorescent images, it is an indicator for cell apoptosis. Therefore, in this work, the MTT assay for test the cytotoxicity to determine cell death overtime has not been performed.
Point 10: Figure 7- the images are very low quality. Clearer images should be provided. Also, images at higher magnification could be helpful.
Response 10: The quality of the images in the Figure 7 has been improved with clearer images.
Point 11: Conclusions: Again, the statement regarding the possible clinical applications of the constructs is not sustained by the data presented in the manuscript.
Response 11:
We want to thank to Reviewer for your comments. The result reveals that in the water or the buffer with pH higher than 7.25 the DOX.HCl gets off DOX.HCl-SiO2 NPs very slowly. We know that the pH values of the blood and physiologic media are around 7.25-7.50. Confocal laser scanning microscopy images showed that the DOX.HCl-SiO2 NPs were better delivered to cancer cells which are more acidic than healthy cells. The results show that the DOX.HCl-SiO2 NPs system is a promising candidate for drug delivery and has potential in clinical applications.
Point 12: References: Very few and outdated references used. Newer data should be referred in the state of the art of the manuscript (articles from 2017-2016 the earliest).
Response 12: We thank to reviewer for this remark, we have been updated with 6 newer references (articles from 2018-2020).
Sincerely,

Round 2
Reviewer 2 Report
The authors have answered all the queries.